# Exciton States and Optical Absorption in CdSe and PbS Nanoplatelets

**DOI:** 10.3390/nano12203690

**Published:** 2022-10-20

**Authors:** Davit A. Baghdasaryan, Volodya A. Harutyunyan, David B. Hayrapetyan, Eduard M. Kazaryan, Sotirios Baskoutas, Hayk A. Sarkisyan

**Affiliations:** 1Institute of Engineering and Physics, Russian-Armenian University, H. Emin 123, Yerevan 0051, Armenia; 2Department of Materials Science, University of Patras, 26504 Patras, Greece; 3Institute of Electronics and Telecommunications, Peter the Great St. Petersburg Polytechnic University, 195251 Saint Petersburg, Russia

**Keywords:** nanoplatelets, size quantization, exciton states, optical absorption

## Abstract

The exciton states and their influence on the optical absorption spectrum of CdSe and PbS nanoplatelets (NPLs) are considered theoretically in this paper. The problem is discussed in cases of strong, intermediate, and weak size quantization regimes of charge carrier motion in NPLs. For each size quantization regime, the corresponding potential that adequately describes the electron–hole interaction in this mode of space quantization of charge carriers is chosen. The single-particle energy spectra and corresponding wave functions for strong intermediate and weak size quantization regimes have been revealed. The dependence of material parameters on the number of monolayers in the sample has been considered. The related selection rules and the dependence of the absorption coefficient on the frequency and polarization direction of the incident light wave were obtained. The interband transition threshold energy dependencies were obtained for each size quantization regime. The effect of dielectric coefficient mismatch and different models of electron–hole interaction potentials have been studied in CdSe and PbS NPLs. It is also shown that with an increase in the linear dimensions of the structure, the threshold frequency of absorption decreases. The binding energies and absorption coefficient results for NPL with different thicknesses agree with the experimental data. The values of the absorption exciton peaks measured experimentally are close to our calculated values for CdSe and PbS samples.

## 1. Introduction

The physical properties of quasi-two-dimensional colloidal semiconductor nanostructures with planar geometry, so-called semiconductor nanoplatelets (NPLs), along with many low-dimensional semiconductors, have also been intensively studied in the last decade (see Refs. [1,2,3,4,5,6,7,8,9,10,11,12,13,14,15,16] and references therein). Along with quantum wells (QWs) and quantum dots (QDs), semiconductor NPLs are the next generation of semiconductor nanostructures with the maximum possible miniaturization of the sample in the growth direction. One of the most intensively and successfully studied objects in this area are semiconductor NPLs of II-VI compounds with wurtzite or zinc blende structure (CdS, CdSe, CdTe [1,2,3,4,5,6,7,8,9,10,11,12,13,14,15,16,17,18,19,20], HgSe, HgTe [21,22]) and, partly, compounds IV–VI (PbS, PbSe, PbTe [4,12,23,24,25], as well as NPLs based on In, Sn, Cu [4]). The dimensions of these systems on the plane of the plate can reach from the tens, hundreds or even thousands of angstroms, while in the transverse direction, the thickness of the NPLs can reach only a few atomic layers and is controlled up to a monolayer precision and almost ideal thickness uniformity [1,2,3,4,5,6,7,8,9,10,11,12,13,14,15,16,17,18,19,20,21,22,23,24,25,26,27,28,29]. Depending on the intrinsic characteristics of the material and the geometric dimensions, these nanostructures can exhibit properties similar to quantum dots (quasi-discrete states of excitons), up to the states where a pronounced charge carriers size quantization anisotropy in the longitudinal and transverse directions is manifested (a continuum of low-dimensional exciton states) [1,2,4,26,27,28,29]. 

The spatial dimensions of the NPLs determine the physics of the states of charge carriers in the sample between the quantum well and the quantum dot. In terms of their properties and possible applications, NPLs differ significantly not only from bulk samples but also from size-quantized structures of the previous generation quantum wells and quantum dots. In the interval between these extreme states, various modes of size quantization will be observed with a rich substructure of exciton states [2,4,8,18,28,29,30].

A medium usually surrounds the grown colloidal NPL with a lower dielectric constant than one of the sample materials. As a result, a strong increase in the exciton binding energy is observed in NPLs compared to the binding energy of a 2D exciton in a quantum well grown on a substrate [16,18,30]. For example, the exciton binding energy in a bulk CdSe sample is on the order of $E_{ex,CdSe}^{3D}=15\;\mathrm{meV}$, in the case of an ordinary quantum well, this energy increases by a factor of 4: $E_{ex,CdSe}^{2D}=60\;\mathrm{meV}$. However, theoretical calculations and experimental results show that in the case of colloidal CdSe NPL, the binding energy of the exciton state at the thickness of $L_{z}=1.21\;\mathrm{nm}$ reaches the values $E_{ex,CdSe}^{NPL}\approx 190\;\mathrm{meV}$ [18]. As a result, a very narrow absorption–emission maximum peak (<10 nm), which is 1/3 of the traditional quantum dot, and an ultrashort radiative fluorescence lifetime (<1 ns) are observed in NPLs [4,16]. At the same time, a significant linear and nonlinear absorption cross-section, giant oscillator strengths, and a low optical amplification threshold are realized in NPLs [2,4,8,11,16,18,30,31]. Therefore, colloidal NPLs are up-and-coming candidates in various fields of electronics and optoelectronics, such as light emitter devices of various ranges, devices for generating and amplifying light, lasers, solar energy harvesting applications, photosensors, photocatalysis, etc. [4,11,21,23,24,31]. The exceptional fluorescent NPLs properties also make them promising candidates for biological and medical applications [14,15,32]. 

One of the most powerful and productive methods for studying the band structure of a semiconductor, and NPLs in particular, is known as the optical method. When looking at the optical properties of NPLs, the main and most important task is determining the energy spectrum of charge carriers, considering the specifics of the manifestation of size quantization effects and exciton states in the nanostructure under consideration. Regarding these studies, there are currently a very large number of experimental and theoretical works (see, for example, [2,3,4,11,12,14,15,16,17,18,19,20,21,22,23,24,25,26,28,31,33,34,35]). The lateral confinement in NPLs leads to the impossibility of the exciton and impurity state motion having an exact analytical description. Therefore, theoretically studied approximate methods for solving such problems can be conditionally divided into two classes: studies using the numerical methods [13,29,36,37,38] and studies using certain model potentials of an electron–hole interaction with the corresponding physical justification in the case of a specific NPL [3,39,40,41,42]. The authors [43,44] reported accurate results on the polarizabilities, absorption spectra, and quasiparticle band gaps in the framework density matrices for the range of nanoclusters. 

In this work, we theoretically consider optical transitions in CdSe and PbS NPLs, respectively, in the regimes of strong, moderate, and weak size quantization of the charge-carrier motion in these structures without accounting for the impurity or defect effects on the NPLs. 

The article’s structure is as follows: in Section 2, the optical absorption in CdSe and PbS NPLs is theoretically calculated in the regimes of strong, moderate, and weak quantization of the motion of charge carriers, and for each case, the necessary description is provided. In Section 3, conclusions are provided regarding the results.

## 2. Optical Transitions in Semiconductor Nanoplatelets

The spatial dimensions of NPLs in directions *X*, *Y*, *Z*, are denoted $L_{x},L_{y},L_{z}$, respectively. Let us first discuss the case of NPL of CdSe; effective masses of charge carriers in the growth direction and the NPL plane are the following: $m_{z}^{c}\equiv m_{⊥}^{c};\;m_{xy}^{c}=m_{yx}^{c}\equiv m_{∥}^{c};\;m_{z}^{v}\equiv m_{⊥}^{v};m_{xy}^{v}=m_{xy}^{v}\equiv m_{∥}^{v};$ Here, index “*c*” refers to the electrons of the conduction band and *v* = *lh*, *hh* to the light and heavy holes in the valence band, respectively. For accuracy, we note that for NPLs, the spin-orbit splitting of the valence band is not considered in the following calculations. We will mainly use the CdSe and PbS materials for numerical estimates as the most typical representatives of the II–VI and IV–VI groups, respectively. It is necessary to consider that specific phenomenon when a spatial localization in the form of size quantization is imposed on systems, the effective masses of light and heavy holes undergo changes and take the following values [43,44]: (1)$$m_{⊥}^{lh}=\frac{m_{0}}{\gamma_{1}+2\gamma_{2}};\;m_{⊥}^{hh}=\frac{m_{0}}{\gamma_{1}-2\gamma_{2}};\;m_{∥}^{lh}=\frac{m_{0}}{\gamma_{1}-\gamma_{2}};\;m_{∥}^{hh}=\frac{m_{0}}{\gamma_{1}+\gamma_{2}}$$

Here $m_{0}$ is the electron mass and $\gamma_{1},\gamma_{2}$ are the parameters of the spherically symmetric Luttinger Hamiltonian [45,46]. Later, we will use the results of ab initio calculations for $\gamma_{1},\gamma_{2}$ [47]. In the calculations, based on the considered physical conditions, more specific adjustments will be made to the values of the effective masses of carriers and other physical characteristics of the materials if necessary. The parameters of the considered materials are presented in Table 1.

Here $d_{0}$ is the atomic monolayer thickness, ML is the number of monolayers, $\mu_{∥}$ is the in-plane reduced effective mass, $E_{g}$ is the band gap, $V_{e}^{0},V_{h}^{0}$ is the depth of the quantum well in the *Z* direction for electrons and holes, respectively, and $\varepsilon ,\varepsilon_{1}$ are the permittivity constants of the material and environment, respectively.

The perturbation associated with the light wave will be presented in the following form [51]:(2)$$\hat{V}=\frac{e}{m_{0}c}\left(\vec{A}\cdot \vec{p}\right);\;\vec{A}\left(\vec{r},t\right)=A_{0}\vec{e}\exp\left[i\left(\vec{q}\cdot \vec{r}-\omega t\right)\right];\;\vec{e}=\vec{e}\left(e_{x},e_{y},e_{z}\right)$$

Here, $A_{0}$ is the amplitude of the incident wave vector potential, $\vec{p}$ is the three-dimensional momentum operator, $\vec{e}$ is the unit polarization vector, $\vec{q},\omega$ are the wave vector and frequency of the photon, respectively, and the *c*-speed of light in the vacuum. It is well known that for the sample under the light-wave perturbation, Equation (2) there are two types of possible transition in quasi-2D systems: interband and intraband. We will discuss both transition mechanisms by considering linear absorption in NPLs. The interband transitions are connected with the electron–hole pair generation or annihilation. For the interband absorption coefficient K(ω), we have the well-known expression [52]:(3)$$K\left(\omega \right)=\sum_{c,v}{\left|M_{c,v}\right|}^{2}\delta \left[\hbar \omega -E_{g}-\sum_{i}\left(E_{c;i}+E_{v;i}\right)\right]$$

Here $E_{g}-$ is the bulk sample bandgap, δ(x) is the Dirac delta function, and $E_{c;i},E_{v;i}$ are the electron and hole energies, respectively. Where the amplitude of the transition is defined by the matrix element $M_{c,v}$:(4)$$M_{c,v}=A_{c,v}\int \Psi_{c,v}\left({\vec{r}}_{c},{\vec{r}}_{v}\right)\delta \left({\vec{r}}_{c}-{\vec{r}}_{v}\right)d{\vec{r}}_{c}d{\vec{r}}_{v}$$

Here, $A_{c,v}$ is the matrix element of the operator in Equation (2), which is calculated on the Bloch amplitudes of valence and conductance bands. For the intersubband transitions in the NPLs, it is well known that such transitions occur when the polarization vector of the input wave is directed along the growth axis of the heterolayer [52]. The absorption coefficient has the following form in this case:(5)$$K\left(\omega \right)=\sum_{p=c,v}\sum_{f,i}{\left|M_{f,i}\right|}^{2}\delta \left[\hbar \omega -\left(E_{i}^{p}+E_{f}^{p}\right)\right]$$

Using the well-known calculation method [52] for the matrix element of the intersubband–intraband transitions from the initial state |i〉 into a final state |f〉, we have the following expression:(6)$$M_{f,i}=-i\frac{\left|e\right|\hbar A_{0}}{m_{0}c}\frac{2}{L_{j}}\int_{0}^{L_{j}}\Psi_{c,v;i}^{\left(0\right)}\left(j\right)\left(e_{j}\frac{d}{dj}\right)\Psi_{c,v;f}^{\left(0\right)}\left(j\right)dj,\;j=x,y,z$$

### 2.1. Strong Quantization Regime

In our case, the quantitative criterion for the implementation of the strong quantization regime is the fulfillment of the following condition:(7)$$L_{x}^{2},L_{y}^{2},L_{z}^{2}\ll {\left(a_{ex}^{3D}\right)}^{2};\;\left(L_{x},L_{y}\geq L_{z}\right)$$

From the energy point of view, condition in Equation (7) means that in the material under consideration, the size quantization energy of charge carriers in any of the *X*, *Y*, and *Z* directions is much greater than the energy of the Coulomb interaction between an electron and a hole [53,54,55]. Approximating the NPL in the *X*, *Y*, *Z* directions by an infinitely deep rectangular potential, neglecting the Coulomb interaction between an electron and a hole, for the envelope wave functions and the energy spectrum of charge carriers in the CdSe NPL in the corresponding directions, we obtain:(8)$$\Psi_{c,v}^{\left(0\right)}\left(r_{i}\right)=\sqrt{\frac{2}{L_{i}}}\sin\frac{\pi n_{i}^{c,v}}{L_{i}}r_{i};\;E_{c,v;i}^{\left(0\right)}=\frac{\pi^{2}\hbar^{2}{\left(n_{i}^{c,v}\right)}^{2}}{2m_{j}^{c,v}L_{i}^{2}},\;i=x,y,z;\;j=∥,\;⊥;\;v=hh,lh\;;n_{i}^{c,v}=1,2,\ldots$$

With the wave functions from Equation (8) for the matrix elements of the interband dipole transitions, we obtain:(9)$$M_{c,v}=A_{c,v}\prod_{i}\int_{0}^{L_{i}}\Psi_{c}^{\left(0\right)}\left(i\right)\Psi_{v}^{\left(0\right)}\left(i\right)di=A_{c,v}\delta_{n_{x}^{c},n_{x}^{v}}\delta_{n_{y}^{c},n_{y}^{v}}\delta_{n_{z}^{c},n_{z}^{v}};\;i=x,y,z$$

Here $\delta_{i,k}$ is the Kronecker symbol. For the threshold frequencies of the interband transitions from the states of light and heavy holes in the valence band to the electronic states of the conduction band, respectively, we have:(10)$$\begin{array}{l} \hbar \omega_{c,v}^{\left(0\right)}=E_{g}+E_{c,tot}^{\left(0\right)}+E_{v,tot}^{\left(0\right)}\;\\ E_{c,tot}^{\left(0\right)}=E_{c,x}^{\left(0\right)}+E_{c,y}^{\left(0\right)}+E_{c,z}^{\left(0\right)}\\ E_{v,tot}^{\left(0\right)}=E_{v,x}^{\left(0\right)}+E_{v,y}^{\left(0\right)}+E_{v,z}^{\left(0\right)}; \end{array}$$

Considering infinitely deep potential well approximation in all three directions X,Y,Z for the threshold frequency of the transition from the states of valence band heavy holes v=hh to the electronic states of the conduction band, we will have:(11)$${\left[\hbar \omega_{c,v}^{\left(0\right)}\right]}_{\min}=E_{g}+\frac{\pi^{2}\hbar^{2}}{2}\left(\frac{1}{m_{∥}^{c}}+\frac{1}{m_{∥}^{hh}}\right)\left(\frac{1}{L_{x}^{2}}+\frac{1}{L_{y}^{2}}\right)+\frac{\pi^{2}\hbar^{2}}{2L_{z}^{2}}\left(\frac{1}{m_{⊥}^{c}}+\frac{1}{m_{⊥}^{hh}}\right).$$

Figure 1a,c show the dependence curves of the threshold energy ${\left[\hbar \omega_{c,v}^{\left(0\right)}\right]}_{\min}$ on the geometric dimensions of the sample calculated using Equation (11) and Figure 1b,d show the same dependence calculated in the framework of the finite potential well. The values of the potential depth are presented in Table 1.

However, the results obtained within the framework of this approximation differ significantly from the results of [42,49]. A more accurate approach that considers the finiteness of the potential well depth in the growth direction allows us to obtain the values of threshold frequency that are more consistent with the known results (see, for example, [42,49]). The size quantization in the growth direction of the NPLs is very strong. In this case, the infinite deep potential approximation is not physically adequate, which is why the order of 1 eV difference between threshold frequencies is calculated by the finite and infinite well models. As can be seen, all the threshold absorption energy curves undergo a decrease with the increase of NPL lateral size. This can be explanted with a decrease in lateral confinement for larger NPLs.

Furthermore, in the case of $L_{x},L_{y}\gg L_{z}$, the particle energy spectrum of NPL can be assumed quasi-continuous. The absorption peak, in this case, is defined mostly by $L_{z}$. The corresponding results agreed with those of [1].

In the case of intraband–intersubband transitions, it is expedient to separately consider the following three cases of linear polarization of the incident wave: (12)$$\vec{e}=\vec{e}\left(1,0,0\right);\;\vec{e}=\vec{e}\left(0,1,0\right);\;\vec{e}=\vec{e}\left(0,0,1\right);$$

For the frequencies of transitions between subbands (*i*, *f*) in the same band and corresponding matrix elements, we obtain, using expression Equation (6), respectively:(13)$$\hbar \omega_{f,i}^{\left(c,v\right)}={\left(E_{c,v;j}^{\left(0\right)}\right)}_{f}-{\left(E_{c,v;j}^{\left(0\right)}\right)}_{i}$$
(14)$$M_{f,i}=-i\frac{\left|e\right|\hbar A_{0}}{m_{0}c}\frac{2}{L_{j}}\frac{{\left(n_{j}^{c,v}\right)}_{f}{\left(n_{j}^{c,v}\right)}_{i}}{{\left(n_{j}^{c,v}\right)}_{f}^{2}-{\left(n_{j}^{c,v}\right)}_{i}^{2}}\delta_{e_{j},1};\;\left(j=x,y,z\right)$$

Consideration of the intraband transitions Equations (13) and (14) under conditions Equation (12) eliminate the mixing of absorption bands in different directions and makes it possible to determine the optical parameters of the sample for each specific size quantization direction.

Let us now turn to the possibility of considering the electrostatic interaction between an electron and a hole during interband transitions when conditions Equation (7) are satisfied in the CdSe and PbS samples in the framework of perturbation theory. The following fact can justify the usage of perturbation theory in this size quantization regime. The typical sizes of NPLs in the *Z*-direction are much smaller than the Bohr radius of the 3D exciton. The perturbation approximation may become adequate in the case of small lateral size. According to the available data in the literature, the radius of the 3D exciton in CdSe is approximately 5–6 nm [56,57,58].

On the other hand, the thickness of the CdSe NPLs is 1–2.5 nm [1,2,27], and the minimum possible dimensions in the plane are about 5–10 nm [1,2,28]. The corresponding characteristics of the PbS NPL differ significantly from those of CdSe. In a bulk sample of PbS, the Bohr radius is 18–20 nm, $E_{ex,PbS}^{3D}=$ 3968 meV [48,55,59] with thicknesses of 1.2–4.6 nm and from several nanometers to the hundreds in lateral size [23,59,60,61,62]. In this case, the exciton effects can be considered within the framework of perturbation theory. The corresponding energy correction, which is the electron–hole interaction potential averaged over the states Equation (8), is presented in the following form:(15)$$\Delta E_{ex}^{\left(1\right)}=-\frac{e^{2}}{\varepsilon }\int_{V}\int_{V}\frac{{\left|\Psi_{c,v}^{\left(0\right)}\left(x_{c,v}\right)\Psi_{c,v}^{\left(0\right)}\left(y_{c,v}\right)\Psi_{c,v}^{\left(0\right)}\left(z_{c,v}\right)\right|}^{2}dx_{c}dy_{c}dz_{c}dx_{v}dy_{v}dz_{v}}{{\left[{\left(x_{c}-x_{v}\right)}^{2}+{\left(y_{c}-y_{v}\right)}^{2}+{\left(z_{c}-z_{v}\right)}^{2}\right]}^{1/2}}$$

Here V is the volume of the NPL system. The numerical calculation of the integral in Equation (15) makes it possible to obtain the values of the energy correction due to the electrostatic interaction between an electron and a hole in the regime of strong quantization in NPL in all directions. Table 2 shows the values of the ratio $\Delta E_{ex}^{\left(1\right)}$ for CdSe and PbS NPLs at various geometric dimensions of the system in the strong quantization regime.

Accounting for the exciton effects in the case of strong quantization leads to a decrease in the threshold frequency from Equation (11) by the following frequency shift:(16)$$\Delta \omega_{c,v}^{\left(0\right)}=\Delta E_{ex}^{\left(1\right)}/\hbar$$

As can be seen from the Table 2, the correction due to the *e*–*h* electrostatic interaction in the case of PbS is several times smaller than in the case of CdSe. This is due to the weaker electron–hole coupling in PbS than in CdSe. It can be seen from Table 2 that the value of the frequency shift Equation (16) decreases with an increase in the size of the system in any of the three directions. This is clear because, with such an increase in size, a decrease in the magnitude of the electrostatic energy of the *e*–*h* interaction from expression Equation (15) occurs.

### 2.2. Intermediate Quantization Mode

In the mode of moderate quantization, it is assumed that the following conditions are satisfied for the geometric dimensions of the NPL [49,50,51]:(17)$$L_{x},L_{y}\sim a_{ex}^{3D};\;L_{z}<<a_{ex}^{3D},L_{x},L_{y}$$

When this condition is met, with sufficient accuracy, we can restrict ourselves to considering the electrostatic interaction between an electron and hole only in the *XY* plane [29,36]. We will consider the effect of the electrostatic interaction on the behavior of charge carriers in NPL within the framework of the numerical variational approximation and represent the trial wave function in the following form [29]:(18)$$\Psi \left({\vec{r}}_{c},{\vec{r}}_{v},\beta \right)=N\left(\beta \right)\Psi_{c,v}^{\left(0\right)}\left(x_{c},x_{v}\right)\Psi_{c,v}^{\left(0\right)}\left(y_{c},y_{v}\right)\Psi_{c,v}^{\left(0\right)}\left(z_{c},z_{v}\right)\exp\left(-\beta \sqrt{{\left(x_{c}-x_{v}\right)}^{2}+{\left(y_{c}-y_{v}\right)}^{2}}\right)$$

Here N(β) is the normalization constant and β is a variation parameter. Therefore, we obtain the following expression for the average value of an electron–hole pair ground state total energy in NPL:(19)$$\begin{array}{c} \langle E_{tot}^{cv}\rangle =\langle E_{kin}^{cv}\rangle +\langle E_{pot}^{cv}\rangle =\frac{\pi^{2}\hbar^{2}}{2\mu_{\left|\right|}^{c,v}L_{x}^{2}}+\frac{\pi^{2}\hbar^{2}}{2\mu_{\left|\right|}^{c,v}L_{y}^{2}}+\frac{\pi^{2}\hbar^{2}}{2\mu_{⊥}^{c,v}L_{z}^{2}}+\frac{\hbar^{2}\beta^{2}}{2\mu_{\left|\right|}^{c,v}}+\\ \int_{V}V_{e-h}\left(\rho \right){\left|\Psi \left(x_{c},x_{v};y_{c},y_{v};z_{c},z;\beta \right)\right|}^{2}dx_{c}dx_{v}dy_{c}dy_{v}dz_{c}dz_{v};\\ \mu_{\left|\right|}^{c,v}=m_{∥}^{c}m_{∥}^{v}/\left(m_{∥}^{c}+m_{∥}^{v}\right);\;\mu_{⊥}^{c,v}=m_{⊥}^{c}m_{⊥}^{v}/\left(m_{⊥}^{c}+m_{⊥}^{v}\right); \end{array}$$
where $V_{e-h}\left(x_{c},x_{v};y_{c},y_{v}\right)$ is the electron and hole attractive potential. From the wave function normalization condition for the normalized factor from Equation (18), we obtain [29,36]:(20)$$N^{2}\left(\beta \right)=\frac{128}{\pi L_{z}^{2}L_{x}L_{y}}{\left[\frac{1}{\beta^{2}}+\frac{\beta }{2{\left(\beta^{2}+\frac{\pi^{2}}{L_{x}^{2}}\right)}^{3/2}}+\frac{\beta }{2{\left(\beta^{2}+\frac{\pi^{2}}{L_{y}^{2}}\right)}^{3/2}}+\frac{\beta }{4{\left(\beta^{2}+\frac{\pi^{2}}{L_{x}^{2}}+\frac{\pi^{2}}{L_{y}^{2}}\right)}^{3/2}}\right]}^{-1}.$$

The total energy minimization condition with respect to the variational parameter has the following form:(21)$$\frac{\partial \langle E_{tot}^{c,v}\rangle }{\partial \beta }=\frac{\partial \langle E_{kin}^{c,v}\rangle }{\partial \beta }+\frac{\partial \langle E_{pot}^{c,v}\rangle }{\partial \beta }=0$$

The following calculations for the $V_{e-h}\left(\rho_{c}-\rho_{v}\right)$ can be chosen in the following forms [29,36,62,63]:(22)$$\begin{array}{l} V_{e-h}^{2D}\left(\rho_{c},\rho_{v}\right)=-\frac{e^{2}}{\varepsilon \sqrt{{\left(x_{e}-x_{v}\right)}^{2}+{\left(y_{e}-y_{v}\right)}^{2}}}=-\frac{e^{2}}{\varepsilon }\frac{1}{\left|\rho_{c}-\rho_{v}\right|}\\ V_{e-h}^{2D}\left(\varepsilon ,\varepsilon_{1},\rho_{c},\rho_{v}\right)=-\sum_{n=-\infty }^{\infty }\frac{q_{n}}{\varepsilon }\int_{0}^{L_{z}}\int_{0}^{L_{z}}\frac{{\left|\Psi_{c}\left(z_{c}\right)\right|}^{2}{\left|\Psi_{v}\left(z_{v}\right)\right|}^{2}dz_{c}dz_{v}}{\sqrt{{\left(\rho_{c}-\rho_{v}\right)}^{2}+{\left[z_{c}-{\left(-1\right)}^{n}z_{c}-nL_{z}\right]}^{2}}};\;q_{n}={\left(\frac{\varepsilon -\varepsilon_{1}}{\varepsilon +\varepsilon_{1}}\right)}^{\left|n\right|} \end{array}$$

Substituting Equations (19), (20) and (22) into Equation (21) and performing numerical calculations of the integral from (19), we obtain the values of the parameter β, satisfying Equation (21). In Figure 2, the graphs of the in-plane wave functions with different positions of the hole depending on the geometrical parameters of CdSe and PbS NPLs are shown.

Figure 2 shows that the influence of lateral confinement is more significant for smaller lateral sizes. This appears because the exciton asymmetry becomes more pronounced in the region closer to the NPL borders. This effect becomes smaller when the hole localization point moves to the NPL center. The values of the corresponding variational parameter β used in the plots above are shown in Figure 3a.

In the case of intermediate quantization, for the threshold energy of the interband transitions, we have:(23)$${\left[\hbar \omega_{c,v}\right]}_{\min}=E_{g}+\langle E_{tot}^{c,v}\rangle =E_{g}+\frac{\hbar^{2}}{2L_{z}^{2}}\left(\frac{1}{m_{⊥}^{e}}+\frac{1}{m_{⊥}^{v}}\right)+\langle E_{kin}^{c,v}\rangle +\langle E_{pot}^{c,v}\rangle$$

Substituting the values of the parameter β into expression Equation (19) and by numerically calculating the integral, it is possible to obtain the corresponding values of the threshold energy of the interband transitions for the different geometric dimensions of the sample (Figure 3b,c). 

Figure 3 and Figure 4 show the results for the moderate size quantization regime for different geometric dimensions of the system in CdSe and PbS NPLs, respectively.

Figure 3 and Figure 4a show the variational parameter β dependence on the NPL lateral size $L_{x}=L_{y}=L$. As can be seen, from the graph, the variational parameter β decreases with further flattening with the lateral size increase. The asymptotic values of the parameter β dependence correspond to the inverse 2D Bohr radius. This is expected as the extremely large lateral sizes trial wave function transforms into the unperturbed 2D exciton wave function. The probability density of the spatial distribution of charge carriers decreases by $e^{-\beta a_{ex}^{2D}}$ compared to the unperturbed spatial distribution. This means that the electron–hole interaction localizes charge carriers in the NPL plane to a smaller region than in the absence of interaction. In addition, the parameter β value decreases with the NPL thickness increase. This can be explained by the fact that the electron–hole interaction becomes weaker with the NPL thickness increase. Figure 3 and Figure 4b,c show the exciton potential energy $\langle E_{pot}^{cv}\rangle$ dependence on NPL lateral size computed in the framework of the variational method using the Coulomb potential (Figure 3 and Figure 4b) and the Takagahara potential (Figure 3 and Figure 4c). The potential energy values using the Takagahara formula Equation (22) are larger than the Coulomb potential case due to the presence of dielectric confinement. In both cases, the exciton potential energy $\langle E_{pot}^{cv}\rangle$ decreases with the increase of NPL lateral sizes. This is related to the fact that the *e*–*h* interaction weakens with the increase of the charge carrier’s localization area. Figure 3 and Figure 4d–f shows threshold energy dependence ${\left[\hbar \omega_{c,v}\right]}_{\min}$ of the interband transitions at various geometric NPL dimensions. In Figure 3 and Figure 4d, the threshold energy was calculated by using $\langle E_{pot}^{cv}\rangle$ computed with the Coulomb potential, Figure 3 and Figure 4e with the Takagahara potential, and Figure 3 and Figure 4f with the Takagahara formula but in the framework of finite potential wells in the growth direction. This shows that the model used in Figure 3 and Figure 4f is the closest to the available literature results [42,63].

Let us now consider the interband absorption coefficient in the NPL in the case of moderate quantization of the carrier motion. For the matrix element of the interband transitions, in this case, we can consider [53]:(24)$$M_{c,v}=A_{c,v}\int \Psi_{c,v}\left({\vec{r}}_{c},{\vec{r}}_{v},\beta \right)\delta \left({\vec{r}}_{c}-{\vec{r}}_{v}\right)d{\vec{r}}_{c}d{\vec{r}}_{v}$$

Substituting now Equations (18), (19), (20) into Equation (24) for the matrix element of the interband transitions, we obtain:(25)$${\left|M_{c,v}\left(\beta \right)\right|}^{2}={\left|A_{c,v}\right|}^{2}\frac{2L_{x}L_{y}}{\pi }{\left|N\left(\beta \right)\right|}^{2}$$

For the absorption coefficient during transitions to the ground exciton state, we obtain, in this case:(26)$$K\left(\omega \right)={\left|M_{c,v}\left(\beta \right)\right|}^{2}\delta \left[\hbar \omega -E_{g}-\frac{\pi^{2}\hbar^{2}}{2L_{z}^{2}}\left(\frac{1}{m_{⊥}^{e}}+\frac{1}{m_{⊥}^{v}}\right)-\langle E_{kin}^{c,v}\rangle -\langle E_{pot}^{c,v}\rangle \right]$$

As can be seen, the intensity of the interband absorption, in this case, is determined by the geometric dimensions of the sample in the *XY* plane. The frequency dependence of the absorption has a resonant character, and the position of the peaks is determined by the threshold energy value ${\left[\hbar \omega_{c,v}\right]}_{\min}$, which in turn is also determined by the geometric dimensions of the sample.

### 2.3. Weak Quantization

In weak quantization mode, it is assumed that the following conditions are satisfied for the geometric dimensions of the NPL [53,54,55]:(27)$$L_{x},L_{y}>>a_{ex}^{3D};\;L_{z}<<a_{ex}^{3D},L_{x},L_{y}$$

When conditions Equation (25) are satisfied, the electrostatic interaction between an electron and a hole, as is known, occurs mainly in the NPL plane [36,42,48,61,62,63]. For the motion of carriers along the Z-axis in the infinite wells model, we can consider the following:(28)$$\Psi \left(z_{e},z_{v}\right)=\sqrt{\frac{2}{L_{z}}}\sin\frac{\pi z_{e}}{L_{z}}\sqrt{\frac{2}{L_{z}}}\sin\frac{\pi z_{v}}{L_{z}};\;E_{⊥}^{ex,conf}=\frac{\pi^{2}\hbar^{2}}{2m_{⊥}^{e}L_{z}^{2}}+\frac{\pi^{2}\hbar^{2}}{2m_{⊥}^{v}L_{z}^{2}}$$

To obtain the wave function and corresponding energy in the growth direction in the model of finite well potential, the Schrödinger equation with such potential also considered the self-potential that occurs due to dielectric constants mismatch is solved numerically. In the regime of weak size quantization, the motion of an electron–hole pair in the NPL plane is separated from the motion of the center of inertia of the bound electron–hole pair and the relative motion of the quasi-two-dimensional hydrogen-like electron–hole formation itself. The exciton states can be considered separately [60,61,64]. The motion of the exciton center of inertia is quantized:(29)$$\Psi_{ex}^{conf}\left(\vec{R}\right)=\sqrt{\frac{2}{L_{x}}}\sin\frac{\pi R_{x}}{L_{x}}\sqrt{\frac{2}{L_{y}}}\sin\frac{\pi R_{y}}{L_{y}};\;\vec{R}=\frac{m_{∥}^{e}{\vec{r}}_{e}+m_{∥}^{v}{\vec{r}}_{v}}{m_{∥}^{e}+m_{∥}^{v}};\;$$
(30)$$E_{∥}^{ex,conf}=\frac{\pi^{2}\hbar^{2}}{2M_{∥}}\left(\frac{1}{L_{x}^{2}}+\frac{1}{L_{y}^{2}}\right);\;M_{∥}=m_{∥}^{e}+m_{∥}^{v};$$

The relative motion is symmetric in the NLP plane; thus, it is possible to further separate the relative motion and consider only the radial part. The radial part of wave functions of the relative motion of the electron and hole can be obtained by numerically solving the Schrodinger equation: (31)$$-\frac{\hbar^{2}}{2\mu_{∥}^{c,v}}\left(\frac{d^{2}\psi \left(\rho \right)}{d\rho^{2}}+\frac{1}{\rho }\frac{d\psi \left(\rho \right)}{d\rho }\right)-\left(E_{ex}^{2D}-V_{e-h}^{2D}\left(\varepsilon ,\varepsilon_{1},\rho \right)\right)\psi \left(\rho \right)$$

Here ε is the permittivity of the colloidal NPL material, a $\varepsilon_{1}$—the permeability of the environment surrounding the nanoplatelet; then, since the environment of 2D materials is often organic materials with significantly lower values of the permittivity, the condition is usually satisfied:(32)$$\varepsilon >\varepsilon_{1}$$

When condition Equation (32) passes into the condition:(33)$$\varepsilon \gg \varepsilon_{1}$$
then the electron and hole in the plane of the NPL interact through the two-dimensional Ritova–Keldish potential [42,48,63,65]. In the case of cadmium selenide, the values available in the literature for this material *ε* = 6–10, strictly speaking, do not allow one to use condition Equation (33) with sufficient accuracy and to calculate the exciton binding energy in NPL in the potential approximation that will be adequate to condition Equation (31). Our calculations unambiguously show that, in the case of CdSe, the choice of the electron–hole interaction potential in the form $V_{e-h}^{2D}=V_{e-h}^{2D}\left(\varepsilon ,\varepsilon_{1},\rho_{c},\rho_{v}\right)$ from Exp. Equation (22) is the most realistic for describing bound exciton states in CdSe NPLs [42,64]. Obtained by averaging over the single-particle states, this two-dimensional potential considers the plate’s real, non-zero thickness along the *Z*-axis, and secondly, the less significant difference between ε and $\varepsilon_{1}$. For example, the calculations performed for the CdSe NPL with ε = 10 and $\varepsilon_{1}$ = 2 with thickness $L_{z}=1.5$ nm (five monolayers) for the exciton binding energy lead to the result $E_{ex}^{2D}=183$ meV. This result agrees quite accurately with the results of a number of works presented in [18,26], where we obtained meV at a thickness of $L_{z}=1.21$ nm (4,5 monolayer) for binding energy. The authors in [42] for the binding energy of the exciton *1S* state, obtained the result $E_{ex}^{2D}=252$ meV for NPL thickness $L_{z}=1.5$ nm and ε = 6, $\varepsilon_{1}$ = 2, which coincides with the data of the corresponding experiment with high accuracy. The calculation using the interaction potential Equation (22) with the same initial data leads to the result $E_{ex}^{2D}=254$ meV. Note that the calculations performed using the potential Equation (31) for the same initial data values lead to the result $E_{ex}^{2D}∼1-1.3\;\mathrm{eV}$. In Table 3, the values of exciton binding energy are $E_{ex}^{2D}$ in the CdSe NPL for different thicknesses $L_{z}$ in the cases ε=6, $\varepsilon_{1}=2$ and ε=10, $\varepsilon_{1}=2$. 

The comparison of calculations at ε=6, $\varepsilon_{1}=2$ shows that the results of calculations using potential Equation (22) are in good agreement with the results of both theoretical and experimental results given in [42]. The exciton radial probability distribution of the *1S* state for different NPL thicknesses is shown in Figure 5. As one can see from the Figure 5, the effective localization radius of the exciton grows with the NPL thickness increase. This is due to the thinner NPL possessing a more significant confinement effect in *Z*-direction, which leads to stronger electron–hole effective potential and, thus smaller effective localization areas. This also can be seen in Table 3. The exciton Bohr radii also grow for thicker NPLs.

Figure 5 demonstrates the results of calculations for the weak size quantization regime. Figure 5a,d show the exciton radial probability distribution of the *1S* state with different NPL thicknesses for CdSe and PbS, respectively. The numerically calculated exciton wave function can be approximated by the *1S* exciton wave function with an estimated exciton Bohr radius $a_{ex}^{2D}=a_{ex}^{2D}\left(L_{z}\right)$ for each value of plate thickness.
(34)$$\psi \left(\rho \right)\approx \frac{2}{a_{ex}^{2D}}\exp\left(-\rho /a_{ex}^{2D}\right)$$

Figure 5b,c,e,f show the absorption coefficient dependence on the incident photon energy for CdSe and PbS, respectively. It can be seen that the absorption coefficient calculated in this size quantization mode is agreed with experimental data for CdSe NPLs [65]. For the case of 4.5 ML (see Figure 5b), there are two excitonic peaks located at 2.450 eV (506 nm) and 2.650 eV (467 nm), although we considered 20 excitonic states with small amplitude and were averaged together. The experimental absorption coefficient, in this case, also has two peaks close to the calculated peaks. For the case of 5.5 ML (Figure 5c), there are also two peaks located at 2.2 eV (563 nm) and 2.32 eV (534 nm), which is also agreed with experimental data, although the is a small difference (<30 meV) between the locations of the 2S peaks. Figure 5e,f show the absorption coefficient dependence on incident photon energy for PbS NPLs for 4 ML and 5 ML, respectively. The results show that there is only one peak for both thicknesses. The peaks located at 1.075 eV (1153 nm) for 4 ML and 930 meV (1333 nm). 

For the matrix element of the interband transition in the case of weak quantization, we can consider [53]: (35)$${\left|M_{c,v}\right|}^{2}={\left|A_{c,v}\right|}^{2}{\left|\psi \left(\rho =0\right)\right|}^{2}{\left|\int \Psi_{ex}^{conf}\left(\vec{R}\right)d\vec{R}\right|}^{2}\approx {\left|A_{c,v}\right|}^{2}\frac{4}{{\left(a_{ex}^{2D}\right)}^{2}}\frac{64L_{x}L_{y}}{\pi^{4}}$$

The following expression gives the threshold frequency of the interband transitions to the final exciton state:(36)$$\hbar \omega_{\min}^{ex}=E_{g}+\frac{\pi^{2}\hbar^{2}}{2\mu_{⊥}^{c,v}L_{z}^{2}}+\frac{\pi^{2}\hbar^{2}}{2M_{∥}}\left(\frac{1}{L_{x}^{2}}+\frac{1}{L_{y}^{2}}\right)-\left|E_{ex}^{2D}\right|$$

Consideration of exciton states in PbS NPL has several features. The binding energy of exciton is sensitive to material parameters. In addition, it is necessary to consider the dependence of the values of the effective mass and other optical characteristics of the sample on the thickness [49]. At the same time, recent studies also show that when calculating the exciton states in the quantum size structures of PbS, it is necessary to use not the static (ε=174) but the optical dielectric permittivity (ε=17) [49]. The theoretical calculations show that the exciton binding energy for PbS increases with decreasing thickness and the dielectric constant of the surrounding material and has values within the range of about 30–100 meV with a thickness between 8 and 2 nm [48,49] see also Table 4. These values are one order higher than what is expected for bulk PbS 5–10 meV, determined by the hydrogen atom approach) [66]. For example, the author considers the dielectric constant to be 17, and the binding energy for a 3 nm thick well is approximately 80 meV [67]. 

In our binding energy calculations considering the dielectric constant mismatch in material and environment and using the Takagahara [68,69] formula of the electron–hole interaction, we obtain 117 meV for 1.8 nm thick NPL with ε=17. This result agreed with the variational Quantum Monte Carlo calculation provided in [48]. A small difference comes from slightly different thicknesses and the presence of lateral confinement in the mentioned paper. When we use the static dielectric constant for exciton binding energy, we obtain 15 meV. The first number is more realistic and agrees with the experimental results; thus, we concluded that ε=17 describes the exciton binding energy better than the static dielectric constant. This fact is illustrated in Table 4, the experimental values of exciton binding energy for 3 and 4 monolayer thick samples 115 meV and 80 meV is close to calculated 93.7 and 77.2 respectively. However, those values are very far from those that calculated with ε=174.

## 3. Conclusions

The results obtained in this work, along with the discussion and comparison with the results of other works, show that the analytical and numerical calculation methods used in the work for each specific size quantization regime allow us to provide an adequate quantitative description of the characteristics of the optical absorption in CdSe and PbS nanoplates. The obtained values of threshold frequencies and exciton binding energies agreed with the experimental data. It has been demonstrated that in the NPLs under consideration, the effective mass and dielectric coefficient strongly depend on the number of sample monolayers and therefore impact the description of single-particle and excitonic states. The suggested approach and corresponding calculations show that in CdSe and PbS NPLs, the naïve usage of the two-dimensional Coulomb potential leads to inaccurate physical results. Therefore, taking into account the essential difference between the dielectric constant of the NPL and surrounding material provides much more adequate results of exciton description in NPLs. In both cases, the exciton binding energy significantly exceeds the values computed in the two-dimensional Colomb interaction model framework. It has been shown that the physical principles and calculation methods used in work can also be used when considering optical transitions in nanoplatelets of other materials of compounds II–VI and IV–VI. In addition, the results can be further used to describe the statistical properties of charge carriers when considering the collective properties of low-dimensional excitons in nanoplatelets.

## Figures and Tables

**Figure 1 nanomaterials-12-03690-f001:**
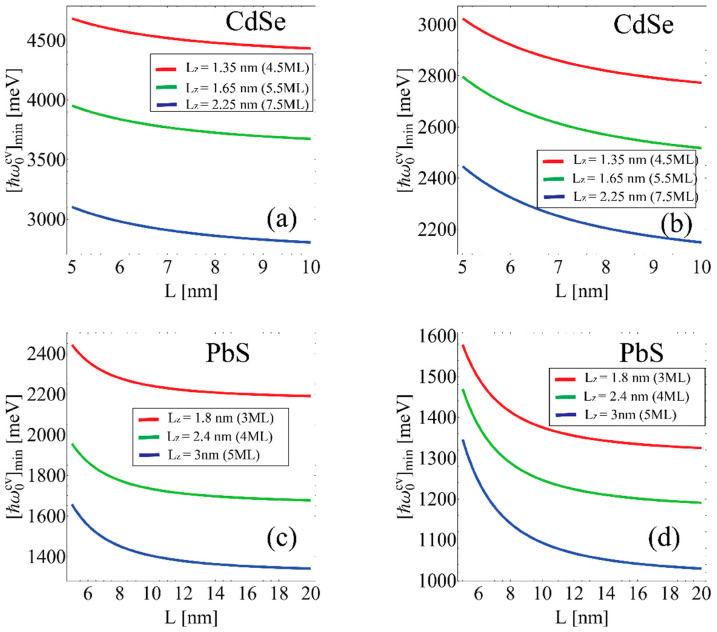
Interband transitions threshold energy dependence ${\left[\hbar \omega_{c,v}^{\left(0\right)}\right]}_{\min}$ in the strong quantization regime on NPL lateral size for different thicknesses. (**a**) CdSe NPL with the impenetrable wall in all directions. (**b**) CdSe NPL with finite barrier height in the z-direction. (**c**) NPL PbS with the impenetrable wall in all directions (**d**) PbS NPL with finite barrier height in z direction.

**Figure 2 nanomaterials-12-03690-f002:**
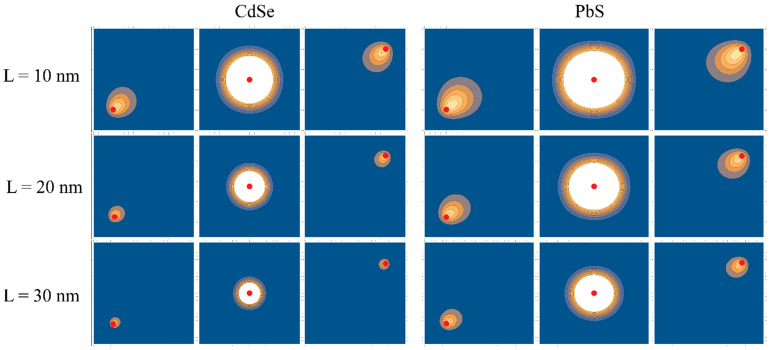
In-plane electron probability density with fixed position of the hole. The left image corresponds to CdSe right one to PbS. The different rows represent different lateral sizes of NPL $L_{x}=L_{y}=L=10,\;20,\;30\;\mathrm{nm}$, respectively. In each column, images represent different hole positions.

**Figure 3 nanomaterials-12-03690-f003:**
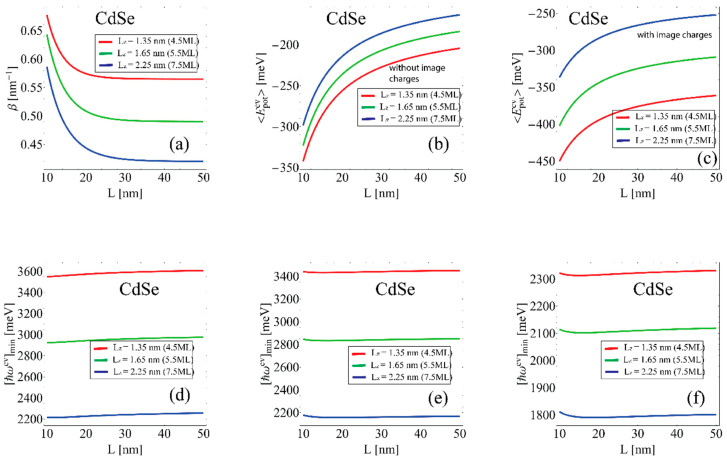
The intermediate regime of size quantization in CdSe NPL. The red, blue and green lines correspond to different NPL thicknesses, 4.5 ML, 5.5 ML, 7.5 ML, in all images, respectively. (**a**) Variational parameter β dependence NPL lateral size $L_{x}=L_{y}=L$. (**b**) Exciton potential energy $\langle E_{pot}^{cv}\rangle$ dependence on NPL lateral size is computed in the variational method framework using the Coulomb potential. (**c**) Exciton potential energy $\langle E_{pot}^{cv}\rangle$ dependence on NPL lateral size computed in the framework of the variational method using the Takagahara potential. (**d**) Threshold energy dependence ${\left[\hbar \omega_{c,v}\right]}_{\min}$ of the interband transitions at various geometric NPL dimensions using $\langle E_{pot}^{cv}\rangle$ computed with the Coulomb potential and the infinitely large walls model in the growth direction. (**e**) Threshold energy dependence ${\left[\hbar \omega_{c,v}\right]}_{\min}$ of the interband transitions at various geometric NPL dimensions using $\langle E_{pot}^{cv}\rangle$ computed with the Takagahara potential and the infinitely large walls model in the growth direction. (**f**) Threshold energy dependence ${\left[\hbar \omega_{c,v}\right]}_{\min}$ of the interband transitions at various geometric NPL dimensions using $\langle E_{pot}^{cv}\rangle$ computed with the Takagahara potential and the finite barrier model in the growth direction.

**Figure 4 nanomaterials-12-03690-f004:**
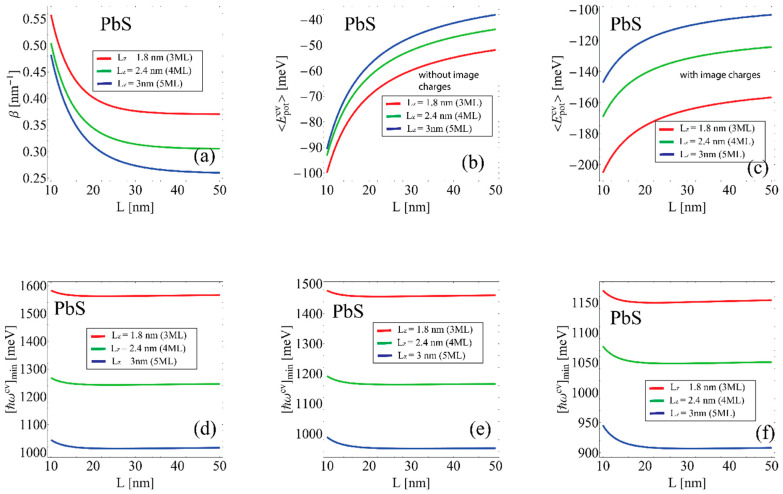
The intermediate regime of size quantization in PbS NPL. The red, blue and green lines correspond to different NPL thicknesses, 3 ML, 4 ML, 5 ML, in all images, respectively. (**a**) Variational parameter β dependence NPL lateral size $L_{x}=L_{y}=L$. (**b**) Exciton potential energy $\langle E_{pot}^{cv}\rangle$ dependence on NPL lateral size, computed in the framework of the variational method using the Coulomb potential. (**c**) exciton potential energy $\langle E_{pot}^{cv}\rangle$ dependence on NPL lateral size, computed in the framework of the variational method using the Takagahara potential. (**d**) Threshold energy dependence ${\left[\hbar \omega_{c,v}\right]}_{\min}$ of the interband transitions at various geometric NPL dimensions using $\langle E_{pot}^{cv}\rangle$ computed with the Coulomb potential and the infinitely large walls model in the growth direction. (**e**) Threshold energy dependence ${\left[\hbar \omega_{c,v}\right]}_{\min}$ of the interband transitions at various geometric NPL dimensions using $\langle E_{pot}^{cv}\rangle$ computed with the Takagahara potential and the infinitely large walls model in the growth direction. (**f**) Threshold energy dependence ${\left[\hbar \omega_{c,v}\right]}_{\min}$ of the interband transitions at various geometric NPL dimensions using $\langle E_{pot}^{cv}\rangle$ computed with the Takagahara potential and the finite barrier model in the growth direction.

**Figure 5 nanomaterials-12-03690-f005:**
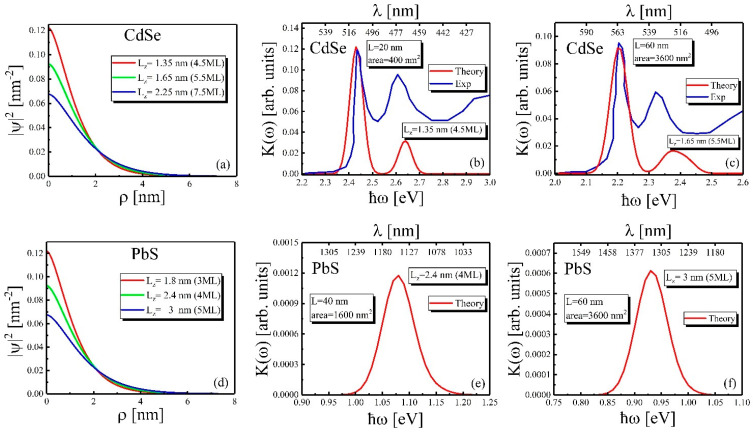
Week size quantization regime in CdSe (first row) and PbS (second row) NPLs. The red, blue and green lines correspond to different NPL thicknesses, 4.5 ML, 5.5 ML, 7.5 ML, for CdSe and 3 ML, 4 ML, 5 ML for PbS in all images. (**a**,**d**) The exciton radial probability distribution of *1S* state with different NPL thicknesses for CdSe and PbS, respectively. (**b**,**c**) The absorption coefficient dependence on incident photon energy for the CdSe NPLs for 4.5 ML and 5.5 ML, respectively. The experimental results are taken from [65]. The peaks widths (about 40 meV) are estimated using the NPL size distribution reported in the same paper. (**e**,**f**) The absorption coefficient dependence on the incident photon energy for the PbS NPLs for 4 ML and 5 ML, respectively.

**Table 1 nanomaterials-12-03690-t001:** Parameters of considered materials. These parameters are taken from [42,48,49,50]. Some of these data points interpolated using available data.

	Material	CdSe	PbS
Parameter			
$d_{0},nm$		0.3	0.598
$m_{⊥}^{e},m_{0}$		0.144 (4.5 ML)	0.29 (3 ML)
		0.138 (5.5 ML)	0.27 (4 ML)
		0.13 (7.5 ML)	0.25 (5 ML)
$m_{⊥}^{hh},m_{0}$		0.92 (4.5 ML)	0.25 (3 ML)
		0.9 (5.5 ML)	0.23 (4 ML)
		0.88 (7.5 ML)	0.21 (5 ML)
$\mu_{∥},m_{0}$		0.09 (4.5 ML)	0.111 (3 ML)
		0.081 (5.5 ML)	0.101 (4 ML)
		0.076 (7.5 ML)	0.089 (5 ML)
$E_{g},eV$		2.15 (4.5 ML)	0.75 (3 ML)
		2.0 (5.5 ML)	0.8 (4 ML)
		1.76 (7.5 ML)	0.74 (5 ML)
$V_{e}^{0},V_{h}^{0},\;eV$		2, 2.5	4, 3
ε		6	17
$\varepsilon_{1}$		2	2

**Table 2 nanomaterials-12-03690-t002:** In the strong quantization regime, the values of perturbation energy correction $\Delta E_{ex}^{\left(1\right)}$ in CdSe and PbS NPLs for various geometric dimensions of the system (CdSe 4.5 ML and 5.5 ML).

$L_{z},\;ML$	4 ML				5 ML			
$L_{x}=L_{y}=L,\;\mathrm{nm}$	4	6	8	10	4	6	8	10
CdSe $\Delta E_{ex}^{\left(1\right)}$, meV	242	170	130	106	233	165	128	104
PbS $\Delta E_{ex}^{\left(1\right)}$, meV	74	54	42	35	70	51	41	34

**Table 3 nanomaterials-12-03690-t003:** The values of the exciton binding energy $E_{ex}^{2D}$ and the Bohr radius $a_{ex}^{2D}$ in the CdSe nanoplate at different layer thicknesses for the cases ε=6, $\varepsilon_{1}=2$ and ε=10, $\varepsilon_{1}=2$. The values of the effective masses of carriers for calculating the reduced mass for various plate thicknesses are taken from [42].

ε	6				10			
$L_{z},\;\mathrm{ML}$	3.5	4.5	5.5	7.5	3.5	4.5	5.5	7.5
$L_{z},\;\mathrm{nm}$	1.05	1.35	1.65	2.25	1.05	1.35	1.65	2.25
$\mu_{∥}^{c,v}$ ($m_{0}$)	0.103	0.09	0.081	0.076	0.103	0.09	0.081	0.076
$E_{1S-1P}^{2D}$ experiment (meV)	214 ± 21	181 ± 17	154 ± 14	120 ± 30	-	-	-	-
$E_{1S-1P}^{2D}$ (meV)	228	187	158	126	158	128	106	84
$E_{ex}^{2D}$ (meV)	313	257	220	180	228	186	158	127
$a_{ex1S}^{2D}$ (nm)	1.5	1.77	2.04	2.38	1.84	2.2	2.54	2.97
$a_{ex1P}^{2D}$ (nm)	6.3	7.41	8.42	9.52	7.328	8.64	9.83	11.3

**Table 4 nanomaterials-12-03690-t004:** Values of the exciton binding energy $E_{ex}^{2D}$ and the Bohr radius $a_{ex}^{2D}$ in the PbS NPL at different layer thicknesses for the cases ε=17, $\varepsilon_{1}=2$ and ε=174, $\varepsilon_{1}=2$.

ε	17				174			
$L_{z}$ (ML)	2	3	4	5	2	3	4	5
$L_{z}$ (nm)	1.2	1,8	2.4	3	1.2	1,8	2.4	3
$\mu_{∥}^{c,v}$ ($m_{0}$)	0.123	0.111	0.101	0.089	0.123	0.111	0.101	0.089
$E_{ex}^{2D}$ (meV)	159.3	117.7	93.7	77.2	20.9	15.3	12.	9.8
$E_{ex}^{2D}$ exp. (meV)	-	-	115	80	-	-	-	-
$E_{1P-1S}^{2D}$ (meV)	59.2	47.1	38.5	32	8.4	6.6	5.4	4.5
$a_{ex1S}^{2D}$ (nm)	2.183	2.754	3.28	3.86	6.409	8.163	9.82	11.66
$a_{ex1P}^{2D}$ (nm)	7.978	9.776	11.59	13.67	21.84	27.1	32.25	38.1

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
