# Peer review of "Exciton States and Optical Absorption in CdSe and PbS Nanoplatelets"

_nanomaterials, 2022, doi:10.3390/nano12203690_

Round 1
Reviewer 1 Report
The manuscript proposes a semi-analytical model for the energy spectra of the electronic subsystem and the absorption spectra of nanostructures in the form of a rectangular parallelepiped. The authors name such objects nanoplatelets (NPLs). Important cases are considered when one of the edges of the parallelepiped has a size of several monolayers (intermediate and weak quantization modes), as well as the case when all three edges of the parallelepiped have sizes of several monolayers (strong quantization mode). In my opinion, the main advantage of the model is that it takes into account the symmetry NPLs and the dependence of their band parameters on the size. Despite the relative simplicity of the proposed model, it gives a quite reasonable description of the experimental absorption spectra, at least in the case of CdSe NPLs. I believe that the proposed article will be of interest to a wide range of researchers working in the field of nanostructure physics. In my opinion, the proposed manuscript can be published in Nanomaterials without changes.
P.S. In further studies, to improve the proposed model, I recommend to consider the spin-orbit splitting of the valence band, that will allow to take into account the non-parabolic nature of the energy spectrum of the NPLs electronic subsystem, as well as introduce three different effective masses for electrons and holes (m_x, m_y, m_z), that, in my opinion, is important, at least in the case of strong quantization mode.
Author Response
We express our gratitude to the reviewers for a detailed review of our manuscript and of useful remarks. The first reviewer did not have comments and was fine with our text.
Reviewer 2 Report
This manuscript reports on the exciton states and optical absorption in CdSe and PbS nano- platelets (NPLs). Not only the proposed theoretical evaluation scheme is detailed and timely developed original approach, but it also comes as both practical and reliable support to experimental results available. The results as presented are very well systematized and convincing. The discussion provided is quite adequate for the present ambitious purpose. The well detailed and at the same time comparative context of the results clarifies convincingly the researched optical absorption in CdSe and PbS NPLs and prompts a good understanding of their excited state properties which can benefit a wide range of developing research dedicated to these material systems.
From practical point of view, the reported results thus bring new knowledge and certainly represent an original contribution in the present context.
The authors chose an adequate structure of the manuscript – an excellent point of departure for such a study. Also, they provided a balanced realistic and nicely illustrated presentation of their results and corresponding analysis that is of much scientific and practical interest and adds new knowledge to the field.
The present manuscript is a significant contribution, this work once published would be instructive and suggestive in terms of further studies and to a wider readership.
There are some minor issues with this already excellent manuscript that will need to be addressed before becoming suitable for publication, i.e., it can be considered for publication after a minor revision:
1: Abstract is well written, but it feels too short and generic. It should be slightly extended reporting 1-2 of the most important benchmarking numerical results. There are other such examples throughout the abstract and the main text too.
2: To certain extent, the above criticism (too short, too generic) applies to the section of the Conclusions. Conclusions should contain the 2-3 most emblematic numerical results and in its finalizing message it should be more focused on possible applications.
3: In the introduction, the authors miss that a wide range of theoretical/modelling/simulation works, including at high but computationally affordable levels of theory have already been heavily adopted/used for studying the excited state/optical/exciton properties of similar low-dimensional nanostructured materials. Such theoretical works that assist understanding structural-excitonic synergies and even may assist experimental work include: The Journal of Physical Chemistry C 118 (2014) 5501-5509 and The Journal of Physical Chemistry C 118 (2014) 11377-11384. Such works should be referenced for achieving a clear picture of broad use of theoretical methods for the purposes as the ones in the present work.
4: Authors should mention, elaborate and be more specific about any role of defects and bonding particularities in the NPLs with the studied chemical composition, namely CdSe and PbS. Defects are known to have impact on the optical absorption of these material systems.
5: Spell-check and stylistic revision of the paper are still necessary. Some, long sentences, misspellings, etc., still are noticeable throughout the text.
Author Response
We express our gratitude to the reviewers for a detailed review of our manuscript and of useful remarks. We made the correction according to reviewer 2.
1: Abstract is well written, but it feels too short and generic. It should be slightly extended reporting 1-2 of the most important benchmarking numerical results. There are other such examples throughout the abstract and the main text too.
The abstract text has been modified according to the reviewer's comment. We added several sentences about numerical results.
2: To certain extent, the above criticism (too short, too generic) applies to the section of the Conclusions. Conclusions should contain the 2-3 most emblematic numerical results and in its finalizing message it should be more focused on possible applications.
The text of the conclusion has been modified and expanded as has been mentioned.
3: In the introduction, the authors miss that a wide range of theoretical/modelling/simulation works, including at high but computationally affordable levels of theory have already been heavily adopted/used for studying the excited state/optical/exciton properties of similar low-dimensional nanostructured materials. Such theoretical works that assist understanding structural-excitonic synergies and even may assist experimental work include: The Journal of Physical Chemistry C 118 (2014) 5501-5509 and The Journal of Physical Chemistry C 118 (2014) 11377-11384. Such works should be referenced for achieving a clear picture of broad use of theoretical methods for the purposes as the ones in the present work.
We added those works in the introduction. Thank you for pointing out those.
4: Authors should mention, elaborate and be more specific about any role of defects and bonding particularities in the NPLs with the studied chemical composition, namely CdSe and PbS. Defects are known to have impact on the optical absorption of these material systems.
The impurity and defects are important for the description of exciton absorption on NPLs, however, in our work, we did not consider such effects. We are thankful to the reviewer for this comment. It is possible that we will consider such a physical situation in the future.
5: Spell-check and stylistic revision of the paper are still necessary. Some, long sentences, misspellings, etc., still are noticeable throughout the text.
We have done stylistic checks.
Thank you,
Hayk Sarkisyan